# Physicochemical and Biological Properties of Mg-Doped Calcium Silicate Endodontic Cement

**DOI:** 10.3390/ma14081843

**Published:** 2021-04-08

**Authors:** Kyung-Hyeon Yoo, Yong-Il Kim, Seog-Young Yoon

**Affiliations:** 1School of Materials Science and Engineering, Pusan National University, Busan 46241, Korea; seweet07@pusan.ac.kr; 2Department of Orthodontics, Dental Research Institute, Pusan National University, Yangsan 50612, Korea

**Keywords:** calcium silicate, endodontic cement, magnesium, ion doping, setting time, hydration, biological properties

## Abstract

Calcium silicate-based cement has been widely used for endodontic repair. However, it has a long setting time and needs to shorten setting time. This study investigated the effects of magnesium (Mg) ion on the setting reaction, mechanical properties, and biological properties of calcium silicate cement (CSC). Sol-gel route was used to synthesize Mg ion-doped calcium silicate cement. Synthesized cement was formulated with the addition of different contents of Mg ion, according to 0, 1, 3, 5 mol% of Mg ion-doped calcium silicate. The synthesized cements were characterized with X-ray diffraction (XRD), Fourier transformed infrared spectroscopy (FT-IR), and scanning electron microscopy (SEM). We also evaluated the physicochemical and biological properties of cement, such as the setting time, compressive strength, micro-hardness, simulated body fluid (SBF) immersion, cytotoxicity, and cell differentiation tests. As a result, the Mg ion improves the hydration properties of calcium silicate cement, and the setting time is reduced by increasing the amounts of Mg ion. However, the mechanical properties deteriorated with increasing Mg ion, and 1 and 3 mol% Mg-doped calcium silicate had appropriate mechanical properties. Also, the results of biological properties such as cytotoxicity, ALP activity, and ARS staining improved with Mg ion. Consequently, the optimal condition is 3 mol% of Mg ion-doped calcium silicate (3%Mg-CSC).

## 1. Introduction

The clinical properties of bioactive endodontic repair cement should be biocompatible, radiopaque, antibacterial, dimensionally stable, easy to hand, and should not be affected by blood contamination [1,2,3,4,5,6]. Among the endodontic cement with favorable properties, mineral trioxide aggregate (MTA) has been widely used. MTA is calcium silicate-based cement (CSC), such as tricalcium silicate (C3S) and dicalcium silicate (C2S), and it contains bismuth oxide for radiopacity. CSC is known as the main product of Portland cement and is responsible for the strength of cement-based materials. However, due to CSC’s handling issues with its long setting time compared to working time (1–30 min) MTA has a long setting time (over 40 min) [7,8]. To overcome the long setting time, CSC (such as Biodentine) with the accelerator was introduced [9,10]. However, the short setting time deteriorates the decrease in mechanical properties [11,12]. Therefore, this study needs to satisfy the ideal clinical properties, especially the setting time and mechanical properties for bioactive endodontic repair cement.

The substitution of trace elements (such as Mg, Sr, Zn, F, etc.) is widely studied to improve the biocompatibility of the calcium silicate ceramics [13,14,15,16,17,18]. Among the various trace elements, magnesium (Mg) is one of the mineral elements that have been studied because of its characteristics having a stimulator influence on the development of bone. It was reported that the incorporation of MgO in the bioactive glass could improve its mechanical properties, biocompatibility, and biodegradation ability [19,20,21,22]. However, Mg-doped calcium silicate cement was not studied for restorative and endodontic usage.

To develop the clinically favorable bioactive endodontic repair cement material, Lee et al. [23] introduced the synthetic process for a sol-gel-derived calcium silicate cement with a short setting time. They demonstrated that the sol-gel process could help the CSC to have clinically favorable mechanical and biological properties [23]. Zhao and Chang also introduced a self-setting pure tricalcium silicate (C3S) with good biocompatibility and surface bioactivity through the sol-gel process [24,25]. The sol-gel method, producing the desired chemical and physical properties using various precursors, is an attractive synthetic process for glass-based materials. It can be easy to adopt an Mg ion with a specific biological role with the desired composition of CSC [26].

This study aims to evaluate the effects of Mg ions on the setting time and mechanical properties of calcium silicate cement. Therefore, we utilized the sol-gel method to synthesize Mg-doped calcium silicate cement to combine the advantages of the sol-gel method and having the efficacy of Mg ion. After the synthesis of Mg-doped CSC, we evaluate the effect of Mg ion on setting time and mechanical properties of calcium silicate cement. Moreover, we investigated the physicochemical and biological properties of these materials for restorative and endodontic usage.

## 2. Materials and Methods

### 2.1. Synthesis of Sol-Gel Calcium Silicate and Mg-Substituted Calcium Silicate

Calcium silicate cement powders were synthesized through the sol-gel process by using Ca(NO_3_)_2_·4H_2_O (Sigma-Aldrich, St. Louis, MO, USA) and Si(OC_2_H_5_)_4_ (TEOS, Sigma-Aldrich, St. Louis, MO, USA) as precursors, and with nitric acid (65%) as a catalyst. The Ca/Si molar ratio of the precursors was set at 3 [23]. Briefly, 65% nitric acid (200 µL) was added to distilled water (15 mL) to catalyze the hydrolysis of TEOS, with stirring for 30 min, and then Ca(NO_3_)_2_·4H_2_O was added to the solution to produce the CSC. To synthesize Mg-CSC (Mg_x_Ca_3−x_SiO_5_) powders, 65% nitric acid (200 µL) was added to distilled water (15 mL) to catalyze the hydrolysis of TEOS, with stirring for 30 min. Then, Mg(NO_3_)_2_·6H_2_O was added with different amounts (1, 3, 5 mol% of Mg ion) to this mixture. After Mg(NO_3_)_2_·6H_2_O was dissolved entirely, Ca(NO_3_)_2_·4H_2_O was added to satisfy the (Mg + Ca)/Si molar ratio of 3. After stirring for 1 h, the obtained solutions were dried at 60 °C for 24 h and 120 °C for 24 h to get gel product, respectively. The dried gel products were calcined at 1400 °C for 2 h [25]. After calcination, the obtained powders were named CSC and 1, 3, 5%Mg-CSC.

### 2.2. Characterization of CSC and Mg-CSC

Crystalline phases of CSC and 1, 3, 5%Mg-CSC powders were determined using an X-ray powder diffractometer (XRD, Xpert 3, Panalytical, Almelo, The Netherlands) with Cu Kα radiation (λ = 0.154 nm), which was generated at 40 kV and 40 mA. Specimen were scanned at a range of 10–60°, and all data were collected in a continuous scan mode at a scanning rate of 3°/min. The chemical bonds of powders were determined using Fourier transform infrared spectroscopy (FT-IR, IS50, Thermofisher Scientific, Waltham, MA, USA) recorded from 400 cm^−1^ to 4000 cm^−1^ with the KBr method (32 scans with 4 cm^−1^ resolution). The particle size and morphology of CSC and 1, 3, 5%Mg-CSC powders were examined, and the Ca/Si ratio of the powders was estimated using scanning electron microscopy (SEM, MIRA3, TESCAN, Männedorf, Switzerland). To investigate the hydration process of calcined powder, we mixed the power with deionized (D. I.) water for 1 day. The hydrated powder also characterized using XRD, FTIR, and SEM.

### 2.3. Setting Time Measurement

For measuring the setting time of CSC and Mg-CSC, cements were mixed with D. I. water (powder to liquid ratio of 2.5:1) for 5 min and placed into a mold (Ø 10 mm). According to ISO 6876, each setting cement indented vertically using flat ends of Gilmore needles to measure initial setting time (diameter of 2.12 mm and a weight of 113.4 g) and final setting time (1.06 mm and a weight of 453.6 g) [27]. The initial setting time was measured when the initial needle started to make an indentation within its height (5 mm). In the case of the final setting time, it was measured until it failed to make the indentation. The measurement was proceeded and repeated every 1 min. Until the indent was not shown, all measurements were repeated five times. 

### 2.4. Compressive Strength Test

After five disks (diameter of 4 mm and height of 6 mm) of each group were prepared and hydrated in the chamber for 1days at 100% humidity, the compressive strength test was proceeded based on the method recommended by ISO 9917-1 [28]. The hydrated specimens examined using a universal testing machine (AGS-X, SHIMADZU, Kyoto, Japan) was used to measure compressive strength at a loading speed of 0.5 mm/min. The maximum compressive load recorded when it was failed to resist the loading force, and compressive strength was calculated as follows:σ = 4F/πD^2^(1)
where σ: compressive strength (MPa), F: the maximum load force before failure (N), D: the diameter of the specimen (mm).

### 2.5. Microhardness Test

The two sets of five disks (diameter of 4 mm and height of 6 mm) from each group were prepared, and each set of the disk was hydrated in the chamber for 24 h with 100% humidity. After hydration, each disk was flattened and polished with SiC paper (600, 800, and 1000 grit) for preparing the microhardness test. The microhardness was tested using a microhardness tester (MVK-H1, Mitutoyo, Kanagawa, Japan). The indents were produced with 98.07 mN load and a dwell time of 6 s was used [23]. After the indentation process, the depth of the indent was measured, and its average was calculated. The average values of depth of indents represent the microhardness of each group. 

### 2.6. Simulated Body Fluid (SBF) Immersion Tests

Bioactivity and hydration products of each group were evaluated by soaking the cement specimens in SBF following ISO 10993-12 [29]. Three test specimens of each group were prepared and stored at 37 °C with 100% humidity for 21 days.

After soaking in SBF for 21 days, the pH values from each group were measured from the groups soaked in SBF solutions for 1, 7, 14, 18, and 21 days. From each group, 1 g of powder was put into test tubes with 10 mL of SBF solutions. The dried specimens were examined using XRD and FT-IR. Differential scanning calorimetry (DSC) was done with a simultaneous thermal analyzer (STA8000, PerkinElmer, Waltham, MA, USA) by a heating specimen from 30 to 600 °C (the heating rate at 10 °C/min) in a nitrogen atmosphere to evaluate the by-product after soaking.

### 2.7. In Vitro Cytotoxicity Test

To test the cytotoxicity of the byproducts of the hydration process, 3-(4,5-dimethylthiazol-2-yl)-2,5-diphenyl tetrazolium bromide (MTT) assay were performed. To evaluate the proper concentration of extract, the material extraction medium was prepared as follows. The specimen was prepared according to ISO 10993-12, extraction method [29]. All specimens were mixed with culture medium at a concentration of 0.1 g/mL to make material extraction medium and incubated for 24 h in 5% CO_2_ at 37 °C. After filtering by a 0.2 μm pore-size syringe filter, the extracted medium solution of each specimen was diluted in DMEM at concentrations of 0%, 12.5%, 25%, 50%, and 100%. Human dental pulp stem cells (hDPSCs) were seeded at a density of 1 × 10^4^ cells/well in a 96-well culture plate. The cells were cultured in the material extraction Dulbecco’s Modified Eagle’s Medium (DMEM; Gibco, Gaithersburg, MD, USA) supplemented with 10% fetal bovine serum (Gibco) and 1% penicillin-streptomycin-neomycin in a 5% CO_2_ incubator at 37 °C for 24, 48 and 72 h.

After 24, 48, and 72 h, the culture medium in each well was replaced with 100 µL fresh culture medium containing 10 µL MTT solution (5 mg/mL MTT in sterile Phosphate-buffered saline (PBS, Sigma-Aldrich, St. Louis, MO, USA) and then incubated for 4 h at 37 °C and 5% CO_2_. Then, the medium was replaced with 100 µL dimethyl sulfoxide (Sigma-Aldrich, St. Louis, MO, USA) to dissolve the formazan crystals for 5 min at 37 °C. The absorbance was subsequently measured at a wavelength of 620 nm using a microplate reader (Sunrise; TECAN, Männedorf, Switzerland). Three independent experiments were performed.

### 2.8. Alkaline Phosphatase (ALP) Activity Assay

Human dental pulp stem cells (hDPSCs) were seeded at a density of 5 × 10^4^ cells/well in a 48-well plate with 25% diluted extraction medium of Mg-CSC (Mg = 0 to 5 mol%) powder [30,31]. hDPSCs were cultured in α-Minimum Essential Medium (α-MEM, Gibco, Grand Island, NY, USA) containing 10 mM β-glycerophosphate, 50 µg/mL ascorbic acid, and 0.1 µM dexamethasone to induce differentiation. The medium was changed every two days. The odontogenic differentiation potential of hDPSCs was evaluated after 7 and 14 days by measuring ALP activity with an ALP Detection Kit (Sigma-Aldrich, St. Louis, MO, USA) according to the manufacturer’s protocol. Briefly, 50 µL ALP substrate solution (1 vial (24 mg) p-nitrophenyl phosphate (pNPP) dissolved in 5 mL ALP buffer containing 0.2 M Tris-HCl, pH 9.5, 1 mM MgCl_2_) were added to 40 µL of cell lysate obtained using an extraction solution and incubated at 37 °C for 60 min. Then, 50 µL stop solution (0.5 N NaOH) was added to each well to stop the reaction, and the absorbance was measured at a wavelength of 405 nm. ALP activity was calculated from the standard calibration curve.

### 2.9. Alizarin Red S Staining and Quantitative Analysis (ARS)

Human dental pulp stem cells (hDPSCs) were seeded at a density of 5 × 10^4^ cells/well in 48-well plates and incubated for 24 h in the medium. The 25% diluted extraction medium was replaced every two days [30,31]. After 21 and 28 days of culture, the cells were rinsed twice with PBS and fixed with 4% paraformaldehyde for 15 min. The specimens were stained with 2% Alizarin red solution for 15 min at room temperature, and the plates were then rinsed three times with DI water, followed by the addition of 10% of cetyl pyridinium chloride for 15 min for quantification. Using a microplate reader (Sunrise; Tecan, Männedorf, Switzerland), the absorbance was measured at a wavelength of 560 nm.

### 2.10. Statistical Analysis

One-way analysis of variance (ANOVA) followed by multiple range tests and Tukey tests were used to test between- Mg-CSC (Mg = 0 to 5 mol%) differences. A *p*-value < 0.05 was considered to indicate a statistically significant difference. All statistical analyses were performed using R (version 3.5.1; R Foundation for Statistical Computing, Vienna, Austria).

## 3. Results

### 3.1. Characterization of Mg-CSC and Hydrated Mg-CSC (Mg = 0 to 5 mol%)

Figure 1 shows the XRD patterns for the Mg-CSC powders with different contents of Mg ion. All kinds of powders were synthesized through the sol-gel method which can produce the desired chemical and physical properties with its homogeneity. To evaluate the structure of Mg-CSC, XRD was performed. Figure 1a,b present the XRD patterns for Mg-CSC powders obtained with calcined at 1400 °C for 2 h. As can be seen in Figure 1a, it seemed to be similar patterns regardless of amount of Mg ion. However, it could be seen that the calcined powders were composed of calcium silicate, especially tricalcium silicate (C3S, ICDD 31-0301) and dicalcium silicate (C2S, ICDD 09-0351) as shown in the enlarged graph (Figure 1b). Except for CSC, the overall Mg-CSC powders (Mg = 1 to 5 mol%) could consist of two phases such as C3S and C2S. Also, in the case of 1, 3, 5%Mg-CSC, the magnesium hydroxide (Mg(OH)_2_, ICDD 44-1482) identified around 18°, 39.5°, and 52°. 

After hydrated for 1 day (as shown in Figure 1c,d), the peak intensity of the C3S phase diminished compared to the C2S phase, and the peak intensity of the C2S phase for all specimens increased. Especially in the CSC specimen, the peaks of the C2S phase (around at 31, 32, 32.6°) were formed. The new peaks corresponding to calcite (CaCO_3_, ICDD 98-016-9922) or magnesian calcite (Ca_1−x_Mg_x_CO_3_, ICDD 01-089-1305) were recorded. The relative intensity of calcite (or magnesian calcite) increased with increasing the contents of Mg ion. For 1, 3, 5% Mg-CSC specimens, the Ca(OH)_2_ peaks were decreased after the hydration reaction.

The FT-IR spectra of un-hydrated and hydrated cement were shown in Figure 2. Since the silicates appear like an intricate band between 1200 and 400 cm^−1^ [23,32] an enlarged graph was shown in Figure 2b. The gray line indicates the common peaks, and the yellow line reveals the peaks generated by Mg substitution.

In the case of calcined powders (Figure 2a), the band between 1407 and 1487 cm^−1^ corresponding to CO_3_^2−^ stretching vibration due to surface adsorption of CO_2_ [33,34,35]. The peak at 3641 cm^−1^ was assigned to O-H stretching vibration [36,37], and the broad absorption bands for H-OH were observed [34]. In Figure 2b, the stretching vibrations of Si-O-Ca appear around 997 cm^−1^ and 872 cm^−1^ in all specimens [23,35,38]. For Mg-CSC (Mg = 1 to 5 mol%), C2S related bands were observed; Si-O asymmetric stretching mode (at 1010, 849 cm^−1^), Si-O bending mode (at 438 cm^−1^), and Si-O-Si out of plane bending mode (at 544 cm^−1^) [35,39]. The absorption band for Si-O stretching was found around 951 cm^−1^ [34]. The O-Si-O and Si-O-Si bending modes occur in the 600–400 cm^−1^ region [23].

The spectra of hydrated Mg-CSC were presented in Figure 2c.d, and the green line is generated bands compared to calcined powder. Overall, a reduction in the intensity of the C3S peak is observed at 997 cm^−1^, 900 cm^−1^, and 872 cm^−1^ [35]. The peaks at 1408 cm^−1^ and 713 cm^−1^ are related to calcite and magnesian calcite (Figure 2c) [11]. The vibration mode of C-O bonding revealed at 1475 cm^−1^ confirming the C-S-H phase, and O-H related bands at 3641 cm^−1^ were disappeared. In hydrated CSC, a broad peak between 1050–800 cm^−1^ indicates the polymerization in the silicate chain of the C-S-H phase [37]. In Figure 2d, the characteristic bands of calcium silicate hydrate (C-S-H) at 980 cm^−1^ assigned to Si-O stretching vibrations [40]. For 3, 5%Mg-CSC specimens, the Si-O-Si bending of C-S-H or magnesium silicate hydrate (M-S-H) appears at 654 cm^−1^ [40]. The Si-O-Mg vibrations revealed at 913 cm^−1^ and 561 cm^−1^ for 1, 3, 5%Mg-CSC specimens [36,41].

Figure 3 illustrated the morphology of Mg-CSC (Mg = 0 to 5 mol%) powders before and after hydration. As can be seen in Figure 3a, calcined powders had small aggregates formed on the surface of calcium silicate particles. In the case of the hydrated test specimen (Figure 3b), all powders presented the hydrated calcium silicate composed of foil-like particles. In the case of 1, 3, 5%Mg-CSC specimens, aragonite-like calcite (magnesian calcite, Ca_1−x_Mg_x_CO_3_) formed, and its size increased as the amount of Mg ion increased.

### 3.2. Setting Time and Mechanical Properties

The Mg-CSC setting times were evaluated using the Gilmore needle test [27], and the results were shown in Figure 4a. The substitution of Mg ion (up to 3 mol%) reduced setting time from 37 to 25 min, and for 5%Mg-CSC, the setting time slightly increased.

Figure 4b,c present the mechanical properties of Mg-CSC after self-setting for 1 day. The compressive strength (as shown in Figure 4b) decreased with the increase of Mg contents from 44 to 7 MPa. Also, in the case of micro-hardness (Figure 4c), it shows the same tendency as the compressive strength. CSC presents the highest value (820 MPa), and the lowest value was found in 5%Mg-CSC specimens (389 MPa).

### 3.3. Characterization of Mg-CSC (Mg = 0 to 5 mol%) after SBF Immersion

The XRD patterns of hydrated products after SBF immersion were shown in Figure 5. It could be seen that all powders were composed of tricalcium silicate (C3S, ICDD 31-0301) and dicalcium silicate (C2S, ICDD 09-0351) in Figure 5a. Also, the hydrated products, such as calcium hydroxide (CH, ICDD 44-1481), magnesium hydroxide (Mg(OH)_2_, ICDD 44-1482), formed after the SBF immersion test. In the enlarged graph (Figure 5b), small calcium-silicate-hydrate (C-S-H, ICDD 3-0247) could be found in all specimens, and the relative intensity of the C3S phase decreased.

Figure 6 shows the FT-IR spectra of Mg-CSC (Mg = 0 to 5 mol%) hydrated products after SBF immersion for 21 days. The two peaks at 3697 cm^−1^ and 3647 cm^−1^ were assigned to Mg(OH)_2_ and Ca(OH)_2_ stretching vibration [42]. The peaks at 3460 and 1647 cm^−1^ were O-H band related to C-S-H and H-O-H vibration bands, respectively [12,33]. In Figure 6b, the green line is generated with bands compared to calcined powder (Figure 2a,b). Same as hydrated powder, the Si-O-Mg vibrations were revealed at 913 cm^−1^, and calcite-related bands were observed at 1408 cm^−1^ and 713 cm^−1^ in the case of Mg-CSC (Mg = 1 to 5 mol%) specimens.

As can be seen in Figure 7a, weight loss is most significant for CSC specimens, and the first weight loss at 60 to 300 °C is related to poorly bound water and dehydration of C-S-H. The endothermic peak around 470 to 550 °C caused by the decomposition of Ca(OH)_2_ [43]. As the amount of Mg ion increases, a new endothermic peak is revealed around 380 °C. This new peak represents to thermal decomposition of Mg(OH)_2_. The SEM images of the Mg-CSC (Mg = 0 to 5 mol%) after the immersion test for 21 days were shown in Figure 7b. It can be seen that all specimens presented the foil-like C-S-H aggregates (yellow arrow) [44,45]. In a 5%Mg-CSC specimen, hydroxyl magnesium silicate (HMS, Mg_6_Si_8_O_20_(OH)_4_)-like mineral also formed (white circle) [46].

### 3.4. In Vitro Cytotoxicity and Differential Test

Cytotoxicity of the Mg-CSC was examined using an MTT assay with different extract concentrations, and results were shown in Figure 8. Overall, the cell viability at 100% and 50% was lower than control (0%), especially for 3% and 5%Mg-CSC. However, at the extracted concentration is 12.5% and 25%, all test groups revealed non-cytotoxicity. The 3, 5%Mg-CSC with a 25% concentration of extract represented the highest cell viability. Especially, the increased cell viability rate is higher at the higher Mg ion contents (3, 5%Mg-CSC).

The differentiation of hDPSCs was tested by the ALP activity and Alizarin S Red using the 25% concentration of extract (in Figure 9). As shown in Figure 9a, ALP activity was measured after three, six, and nine days of cell culture. The control group is tested using a non-extracted medium. In three days of ALP activity analysis, 1% and 3%Mg-CSC groups increase. There were no statistically significant differences between groups for six days. However, after nine days, Mg-CSC showed a statistically significant increase compared to control (medium).

Mineralization related to differentiation of hDPSCs was shown in Figure 9b and c. The quantitative results of ARS staining were shown in Figure 9b. The 3, 5%Mg-CSC showed a statistically significant increase at 14 and 21 days with concentration-dependently. Also, in Figure 9c, the higher Mg ion-doped specimen had a higher mineralization area against the low Mg contained specimens (CSC and 1%Mg-CSC).

## 4. Discussion

A long setting time is one of the biggest problems in calcium silicate cement (CSC). Because the setting time is related to the hydration process, it needs to understand the cement’s hydration reaction. Interestingly, setting time and mechanical properties present the opposite tendencies [11,12]. Therefore, it is essential to find the optimal condition of calcium silicate-based cement. In this study, we investigate the effect of Mg ion on the setting time, mechanical, and biological properties of CSC. The Mg-CSC (Mg = 0–5 mol%) were synthesized by the sol-gel method with the (Ca + Mg)/Si ratio of 3. The crystallinity of synthesized powders was observed with XRD, and all powder presents the C3S and C2S phases (Figure 1a,b). As the amount of Mg ion increases, the C3S peaks shift to a higher diffraction angle. These peak shift towards higher 2θ indicates a reducing d-spacing because the ionic radius of Mg^2+^ (0.74 Å) is smaller than Ca^2+^ (1.04 Å) [47]. Also, in Mg-CSC (Mg = 1 to 5%) specimens, the C2S peaks were formed, and these peaks present no shift with different contents of Mg ion. In FT-IR results, the Si-O-Ca vibrations related C3S phase presented in all specimens [23,35,38]. The absorption bands also indicated the formation of the C2S phase at 1010 cm^−1^, 849 cm^−1^, and 438 cm^−1^ in the FT-IR results (Figure 2a,b) [35,39]. We observed Si-O-Mg vibrations at 913 cm^−1^ (Figure 1c,d) [36]. It could be seen that the Mg ion was inserted into the calcium silicate structure. These results show the Mg ion incorporated into the C3S phase. Also, because the Ca/Si ratio decrease with increasing the Mg ion, the C2S phase is synthesized, and residual Mg ion remained in the Mg(OH)_2_ phase (Figure 1a,b and Figure 2a,b) [42].

After hydration of calcium silicate cement, various by-products were formed according to the hydration duration. Although the hydrated by-products are the same in both C3S and C2S, different reaction processes were provided [33,35,44,48].

The hydration reaction of the C3S and C2S represents dissolution with Ca(OH)_2_ and calcium silicate hydrate (CSH) production. The expected hydration reactions are as follows [12,49,50]: Ca_3_SiO_5_ + nH_2_O → x(CaO)·SiO_2_·(n + x − 3)(H_2_O) + (3 − x)Ca(OH)_2_(2)
Ca_2_SiO_4_ + 4.3H_2_O → 0.3Ca(OH)_2_ + (CaO)_1.7_SiO_2_(H_2_O)_4_(3)

The CaCO_3_ is derived from the surface carbonation of the CH and C-S-H with CO_2_, and these reactions were revealed at Equations (4) and (5) [48]. According to these processes, calcium carbonate is the last by-product during hydration.
Ca(OH)_2_ + CO_2_ + nH_2_O → CaCO_3_ + (n + 1)H_2_O(4)
C-S-H + CO_2_ + H_2_O → CaCO_3_ + SiO_2_·2H_2_O(5)

Because the C3S reacts more rapidly than C2S, during the initial hydration process of CSC, the C3S peaks diminished compared to C2S peaks in Figure 1d. The hydrated calcium silicate (C-S-H) phase was not discernible in XRD results because it was an amorphous phase [51]. For the CSC specimen, the main peak of C3S more shifts to a higher angle (32.1°→32.2°) than Mg-CSC (almost 0.03° shifts). The peaks shift toward a higher angle mean decreasing the d-spacing after hydration and dissolution. Therefore, the C3S phase in the CSC specimen hydrated faster than other specimens. These results show the Mg ion stabilized the C3S phase. However, the hydrated by-products were more formed in Mg-CSC specimens than CSC [40]. As can be found in XRD results (Figure 1c,d), the relative intensity of by-product-related peaks was strong in Mg-CSC specimens. Also, the various O-Si-O stretching bands broaden with the increasing polymerization of the silicate chain of the C-S-H phase [49]. The loss of the OH group is associated with the formation of MgCO3 and the decomposition of Si-OH [37]. Therefore, regardless of the hydration rate of C3S, hydrated by-products were more formed at Mg-CSC specimens.

The setting property of calcium silicate cement is related to the hydration process. Therefore, fast hydration might lead to a short setting time. Our results (in Figure 4a) found that overall specimens have a short setting time than MTA, set after 3 h [52]. Also, as the amount of Mg ion increase, the setting time decreased remarkably. Acceleration of the hydration reaction of C3S is related to the generation of the active site by the incorporation of foreign oxides or other active phases [34,50,53]. In this study, Mg ion contained powder had Mg-related by-products, and these by-products might play a role in setting accelerator. Therefore, these minor components of Mg accelerate the hydration process of calcium silicate cement. In view of the calcium carbonate precipitation, MgO absorbs CO_2_ from the atmosphere [54], Mg incorporated into the calcite and formed magnesian calcite(Ca_1−x_Mg_x_CO_3_). Also, Mg ion reduced calcite precipitation and increased aragonite (a kind of reactive calcium carbonate) fraction [55]. Incorporate with our results, Mg ion accelerates the overall hydration of cement and formed magnesian calcite. Formation of magnesian calcite could observe in XRD (Figure 1c,d; at 2 theta = 29.4 and 34.5°), CO_3_^−^ vibration in FT-IR (Figure 2c,d) [11], and micro-rod particles in SEM images (Figure 3b). 

Mechanical properties show the opposite tendency at setting time. In self-setting progress, the initial formed amorphous CSH decreased the mechanical properties. As time proceeds, amorphous CSH gel polymerized and hardened on the calcium silicate and CH. It is associated with densification and an increase in mechanical strength [24,33]. Also, because the CH has a lower molar volume, the formation of CaCO_3_ (which has a higher molar volume) hardened the cement [45,48]. Thus, the mechanical properties of cement getting increased with hydration time [45]. Incorporate with our results, the similarity of compressive strength between 1%Mg-CSC and 3%Mg-CSC is the existence of calcium carbonate. It is also expected to a higher compressive strength of Mg-CSC (Mg = 0%–5%) after 1 day. Microhardness also decreased with the increase in the amount of Mg ions. However, the lowest value for 5%Mg-CSC (389 MPa) was similar to commercial cement especially for gray MTA (GMTA, 275 MPa) and white MTA (WMTA, 240 MPa) [23]. 

The results of the SBF immersion test were different from the hydration test. The intensive CH peaks were found in XRD results (Figure 5). This result is because the phosphate inhibits calcite formation; only phosphate and sulfate might intervene with the crystallization of calcium carbonate in SBF [56,57]. Furthermore, in 5%Mg-CSC, Mg(OH)_2_ found in the DCS curve (Figure 7a) [43], and hydroxyl magnesium silicate could identify in SEM images (Figure 7b) [46]. According to these Mg-related minor components, Mg ions are involved in the formation of calcium carbonate during hydration.

Determination of cytotoxicity for bioactive materials needs to have a pre-conditioning process [58] or find the optimal concentration of extract [31,59]. Therefore, in this study, the MTT assay of hDPSCs was cultured in the five different concentrations of Mg-CSC extracts (100, 50, 25, 12.5, 0%). As a result, the MTT assay revealed that the higher concentrations (100% extracts) had lower cytotoxicity. This is because of the release of Ca(OH)_2_ during the hydration process [33]. Therefore, the drastically dropped in 100% concentration of 3, 5%Mg-CSC related to its fast-hydration and setting. Also, for 3, 5%Mg-CSC specimens, the cell viability increased with decreasing in the concentration of extract (100% → 25%). This result means the Mg-doped CSC has the optimal concentration to use in clinical applications and in this study, 25% extract is the optimal condition.

To identify the effect of Mg doping on the differentiation of hDPSCs, the ALP activity and ARS staining were investigated using a 25% concentration of extract (Figure 9). Mg-CSC showed an increase in ALP activity after nine days, and ARS staining after 14 and 21 days, respectively. This is believed that Mg-doping upregulates the ALP expression and promotes osteogenic differentiation of hDPSCs. It is consistent with Zhang et al.’s study, which reported the Mg ion could accelerate the osteogenic differentiation by regulating the expression of gene and protein in various stages of mesenchymal stem cell’s osteogenic differentiation [60]. Considering all results from cell test (MTT assay, ALP activity, ARS staining), the CSC and Mg-CSC (Mg = 1 to 5 mol%) specimens would be biocompatible to the hDPSCs.

The relation between the setting time and compressive strength was shown in Figure 10. All specimens have a short setting time than MTA. Compared with previous results using hydration accelerator (NaH_2_PO_4_) [11,61] or curing liquid (K_2_HPO_4_) [12], the setting time of Mg-CSC (Mg = 0–5 mol%) has a lower or similar value without the addition of accelerator [23,54,62]. Accordingly, it is expected that the setting time could be further reduced with the use of additives.

The results of cytotoxicity are much affected by the concentration of extract because the CSC is bioactive material before it hardens. Although we decided the optimal concentration is 25% extract, the biological behaviors in the actual environment are unexpected. Therefore, further in vivo cytotoxicity tests for bioactive materials would be necessary to define the optimal concentration for clinical applications. 

## 5. Conclusions

In this study, the Mg-doped calcium silicate materials were successfully synthesized by the sol-gel method. The Mg-containing CSC materials also have the Mg-related minor component, such as Mg(OH)_2_. These minor components play the role of hydration accelerator and the setting time is shortened. As the Mg ion contents increase, the mechanical properties decreased. The cytotoxicity test showed that the concentration of Mg-CSC (Mg = 0 to 5 mol%) affects cell viability. In this study, the optimal concentration is 25% extract. Regarding the results of the differentiation cell test using 25% extract, the differentiation of hDPSCs is accelerated with an increase in the contents of Mg ions. This is because the Mg ion accelerates the osteogenic differentiation. Considering these results comprehensively, 3 mol% of Mg ion-containing calcium silicate is optimal for endodontic cement. However, the concentration of extract exceeds 50% has cytotoxicity. Therefore, for bioactive materials like cement, it is important to identify the optimal concentration to apply the clinical use.

## Figures and Tables

**Figure 1 materials-14-01843-f001:**
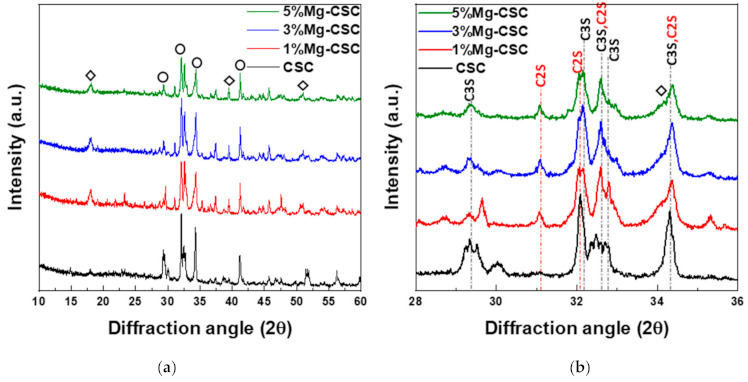
XRD patterns for the Mg-CSC powder with different contents of Mg ion (Mg = 0 to 5 mol%); as a calcined powder (**a**) 2θ: 10–60°; (**b**) 2θ: 25–45°; and hydrated powder (**c**) 2θ: 10–60°; (**d**) 2θ: 27.5–40° (◇: Mg(OH)_2_, ○: tricalcium silicate (C3S) or dicalcium silicate (C2S), △: CaCO_3_ or Ca_1−x_Mg_x_CO_3_).

**Figure 2 materials-14-01843-f002:**
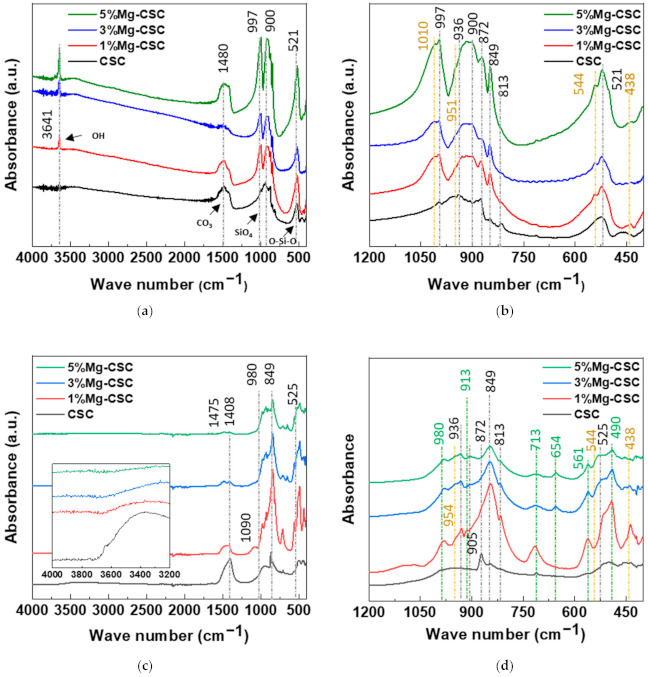
FT-IR spectra of Mg-CSC powder; as calcined (**a**) 4000–400 cm^−1^; (**b**) 1200–400 cm^−1^; and hydrated cement (**c**) 4000–400 cm^−1^; (**d**) 1200–400 cm^−1^.

**Figure 3 materials-14-01843-f003:**
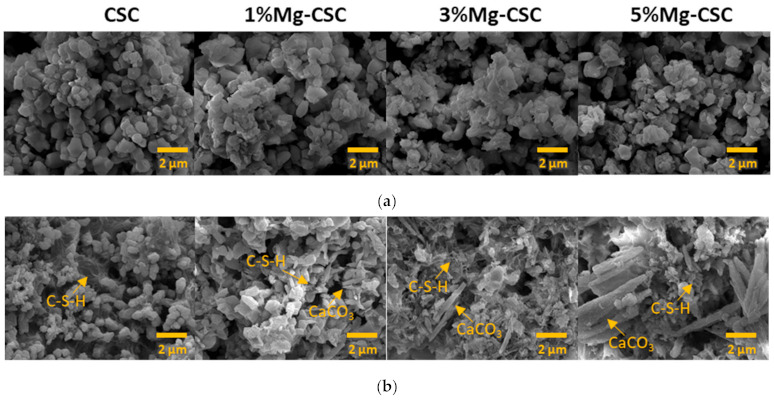
SEM images of Mg-CSC specimens; (**a**) before; and (**b**) after hydration for 1 day.

**Figure 4 materials-14-01843-f004:**
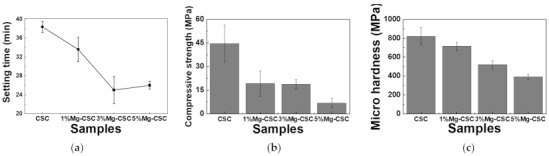
(**a**) Setting times of Mg-CSC (Mg = 0 to 5 mol%) and mechanical properties of Mg-CSC specimens after setting for 1 day; (**b**) compressive strength and (**c**) micro-hardness. Error bars indicate the ± standard deviation.

**Figure 5 materials-14-01843-f005:**
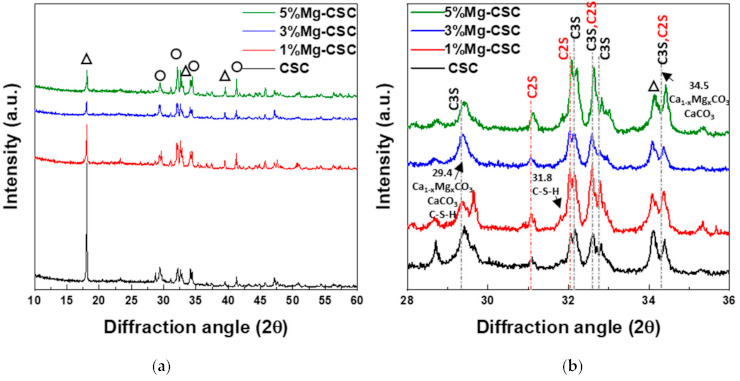
XRD patterns for the Mg-CSC hydrated products after SBF immersion for 21 days; (**a**) 2θ: 10–60° and (**b**) 28–36° (△: Mg(OH)_2_ or Ca(OH)_2_, ○: C2S or C3S).

**Figure 6 materials-14-01843-f006:**
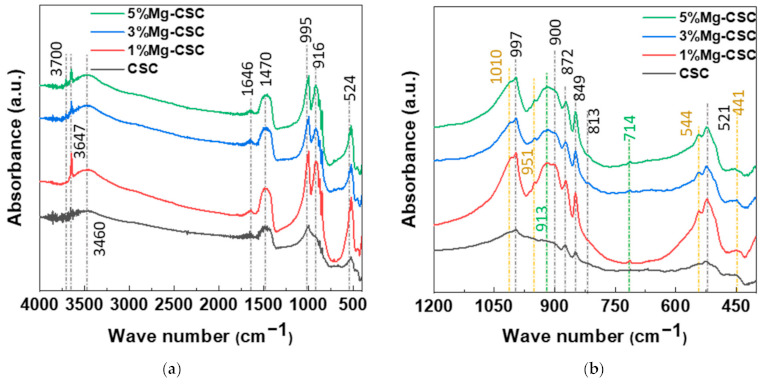
FTIR spectra of Mg-CSC after SBF immersion after 21days; (**a**) 4000–400 cm^−1^; (**b**) 1200–400 cm^−1^.

**Figure 7 materials-14-01843-f007:**
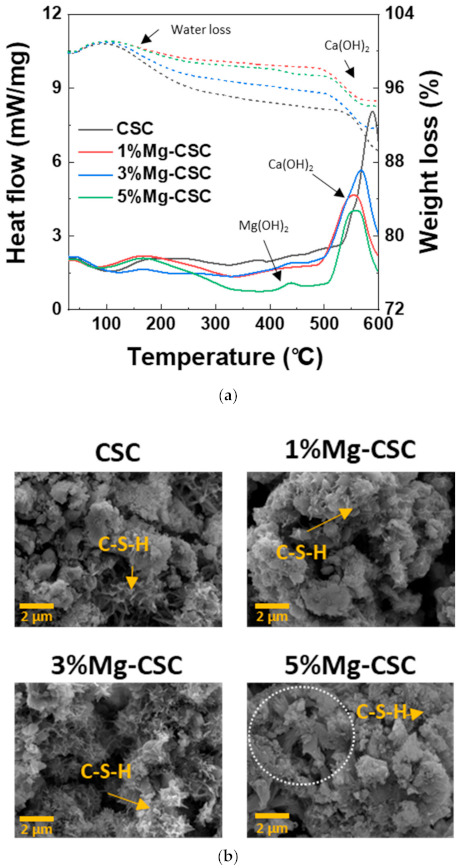
(**a**) DSC curves; and (**b**) SEM images for the Mg-CSC (Mg = 0 to 5 mol%) after SBF immersion test for 21 days.

**Figure 8 materials-14-01843-f008:**
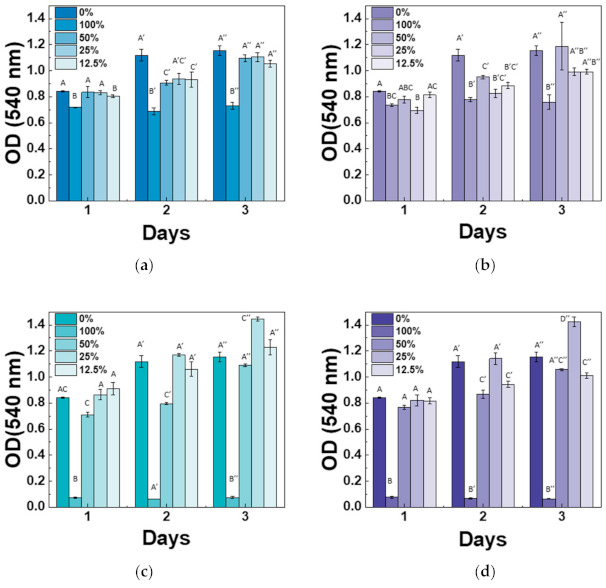
Results of cytotoxicity tests for (**a**) CSC (control); (**b**) 1%Mg-CSC; (**c**) 3%Mg-CSC; (**d**) 5%Mg-CSC. ANOVA was performed. The different letters indicate a significant difference (*p* < 0.05) between groups. Error bars indicate the ± standard deviation.

**Figure 9 materials-14-01843-f009:**
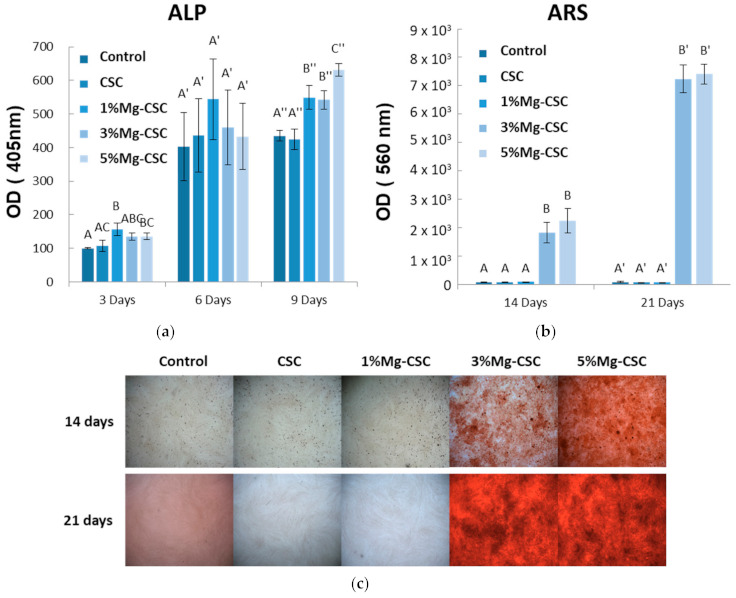
The results of hDPSCs differential test; (**a**) Alkaline Phosphatase (ALP) activity assay; Alizarin Red S (**b**) Quantitative; and (**c**) Staining Analysis (ARS). ANOVA was performed. The different letters indicate a significant difference (*p* < 0.05) between groups. Error bars indicate the ± standard deviation.

**Figure 10 materials-14-01843-f010:**
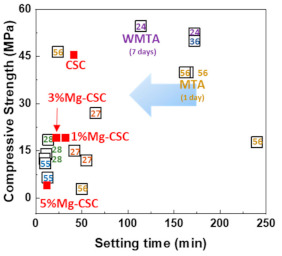
The relation between the setting time and compressive strength (number: reference data) [11,12,23,33,61,62].

## Data Availability

Data sharing is not applicable to this article.

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
