# Peer review of "Physicochemical and Biological Properties of Mg-Doped Calcium Silicate Endodontic Cement"

_materials, 2021, doi:10.3390/ma14081843_

Round 1

Reviewer 1 Report

Introduction. (1) It would be worthwhile in the first paragraph to provide approximate values in minutes for the setting times being discussed, along with a clinically optimum setting time for a bioactive endodontic root repair cement. (2) Brief comments about how the setting time is determined by the clinical endodontist vs. laboratory measurement techniques would also be worthwhile. (3) Many readers will not know what is meant by “C3S”; the compound name should be noted here parenthetically; it is defined later in the Results section.

Materials and Methods. (1) Can a reference be provided for the detailed synthesis protocol described in Section 2.1? (2) This reviewer encourages the authors to use “test specimen” or “specimen”, rather than “sample”; in statistics the latter term describes a group of nominally identical replicate test specimens. (3) What load was employed for the microhardness test? Presumably, a Vickers test was utilized, from information presented later in the Results section. (4) It would be worthwhile to provide references for the detailed biological testing protocols presented in Section 2.8 (ALP activity assay) and Section 2.9 (ARS staining and quantitative analysis). (5) Were power analyses initially performed on preliminary data to determine the sample sizes for the different experiments that would be needed to demonstrate statistical significance (or non-significance)?

Results. (1) Was there evidence of preferred crystallographic orientation of the ceramic phases in the XRD patterns, or were the relative peaks intensities for the phases consistent with those for the corresponding ICDD pure powder standards? (2) Do the XRD patterns indicate formation of the reaction gel phase, perhaps from the increased background at the lowest diffraction angles? (3) Can a reference or references be provided in this section to support the interpretations of the FTIR peaks? (4) In Figure 4 (a) – (c), do the range bars show the standard deviations? (5) Can references be provided to support the stated thermal decomposition interpretations for DSC peaks? (6) More information needs to be provided about the symbols shown on the bar graphs in Figures 8 and 9. Are standard deviations or standard errors being shown with the mean values?

Discussion. (1) Be careful that all of the subscripts are being shown for the chemical formulas of compounds. (2) There should be some comments about the sensitivity of the traditional Gillmore needle modality for determination of the setting time. (3) A concise additional paragraph is recommended that summarizes the limitations of the study and provides suggestions for future research.

Reviewer 2 Report

This study investigated the effects of magnesium (Mg) ion on the setting reaction, mechanical properties, and biological properties of calcium silicate cements (CSC). Generally speaking, this is a very interesting paper which can be of general interest to the readers of Materials. The conclusions are generally supported by the experiments and the paper is well-organized. However, there are some problems the authors may look into.

  1. The introduction can include more contents. calcium silicate cement is more often used as construction materials (etc. Ordinary Portland cement). The binding phase is so-called calcium silicate hydrate (C-S-H). (e.g., "Comparison of the physical and mechanical properties of MTA and Portland cement. Journal of endodontics32(3), pp.193-197."; "Analytical investigation of phase assemblages of alkali-activated materials in CaO-SiO2-Al2O3 systems: The management of reaction products and designing of precursors. Materials & Design194, p.108975." etc.) what is the major difference between the cement for construction and the CSC used as endodontic repair cement?
  2. The last paragraph of Introduction should be rewritten. What is the limitations of the current understanding of the problem? What is the innovation of this paper? The authors should also explain the objective of this research and the major experimental methods.
  3. Do you have any references for the section of "Synthesis of sol-gel calcium silicate and Mg-substituted calcium silicate"? why did you use this method? Why the dried gel products were calcined at 1400 °C for 2 h?
  4. The dimension of samples for compressive strength test should be specified.
  5. The hydration products may also include magnesium silicate hydrate (M-S-H), although it might be generally amorphous and hard to be detected by XRD. Please check the reference paper: "Analytical investigation of phase assemblages of alkali-activated materials in CaO-SiO2-Al2O3 systems: The management of reaction products and designing of precursors. Materials & Design194, p.108975."
  6. The detection of CaCO3 and MgCaCO3 might be because of the carbonation. The samples were contaminated by the CO2 in air.
  7. The conclusion part is too simple. Please give more detailed description of the experimental results, the role of Mg and the possible explanations.
  8. What is the limitation of this research? what may prevent this technology from being widely used?

This is a very interesting paper and I think it should be published after some revisions.

Reviewer 3 Report

The reviewed article deals with the setting reaction, mechanical properties, and biological properties of calcium silicate cements modified with magnesium ions. This material shows many promising properties, however the article requires some adjustments.

Major comments

  1. The scientific goal was not defined in the introduction
  2. SI units are preferable
  3. Line 72: The authors provide the composition of the reactants, but the actual chemical composition of the obtained products has not been verified
  4. 114: (…) set of the disk was hydrated for 24 hours with 100 % humidity (…), and next - were the samples incubated in an environment with a relative humidity of 100% or immersed in a liquid?
  5. 116: The indentation force in microhardness is missing. Units of microhardness measurements are incorrect (Chapter 3.2)
  6. I have the most serious objections about cytotoxicity tests. The ISO standard precisely defines cytotoxicity, results of such tests (viability) were not presented in the article. Moreover, the extracts are used only to determine the IC50 value, defining the concentration of the extract causing a toxic effect on 50% of the cell population. On the other hand, it is obvious that the released ions will show a strong cytotoxicity - what prompted the authors to carry out such studies? The explanation of control samples is also missing.
  7. 286: “… peak of Ca(OH)2 had the highest intensity than other samples” - what is such a high peak intensity related to? what does it matter for the test results?
  8. Fig 9a – no explanations of symbols A, AC, B etc. are given
  9. The first paragraph of the discussion essentially repeats the introduction
  10. 366 and 368 and next – incomprehensible citations - do they confirm the authors' theses, or do the authors conclude something based on them?
  11. 382: “These Mg(OH)2 also can be confirmed in the SEM image…” - is the specific property of Mg(OH)2 to form agglomerates and have they been identified on this basis? Rather, it is suggested that this should be confirmed by the EDS analysis.
  12. In the Discussion section, the word "previous" is often repeated, suggesting the authors' previous research. However, these are references to other works. Linguistic correction is strongly recommended
  13. In conclusions authors stated, that “biological properties are enhanced”. Regarding the strong cytotoxicity, the proposed modification will not be advantageous. I have similar objections about the "optimality" statement for endodontic applications.

Minor comments

  1. Line 29: “Clinical properties of bioactive endodontic repair cements should be biocompatible…” the properties appear to be biocompatible ?
  2. 33: residual toxic elements in MTA – explain in details, which elements
  3. 39: C3S and then C2S - abbreviations are not explained
  4. 43: Besides, the substitution of particular trace elements such as Mg, Sr, Zn, F, etc. (…) - vaguely, it is not known whether you want to replace these elements, or if these elements are to replace others
  5. 49: Also, Mg can enhance the bioactivity of CSC endodontic repair cement - statement or supposition?
  6. 52: Lee et al. […].
  7. 53: “They demonstrated … “ rather than “It demonstrated (…)”
  8. 87: abbreviation D.I. is not explained
  9. It is suggested to provide only the ICDD card number (including full label) for the substances identified in the XRD studies
  10. Correct symbols in Fig 1 caption
  11. 3 and 7a – SEM images are too small
  12. Incorrect writing of chemical formulas - no subscripts
  13. 380: “CH” is not explained

Round 2

Reviewer 1 Report

Thank you for conscientiously addressing the comments of this reviewer for the original manuscript version.

Author Response

We, the authors checked the English once again. Also, we deeply appreciate and checked the English once again.

Reviewer 2 Report

This paper has been revised significantly and should be ready for publication.

Author Response

(The authors gave the same response as above.)

Reviewer 3 Report

The authors made many corrections, however I still have a few comments

  1. The purpose of the work still looks like those in the technical reports instead of a research paper. Moreover, the “optimal properties for endodontics usage” (line 59) have not been defined.
  2. Microhardness units. The main Author (K.-H.Y) works in the School of Materials Science and Engineering. In my opinion, the knowledge of the basic units in the field of material research should be natural. It turns out, however, that there are some problems with this. Please refer to the relevant standard and use the appropriate units.
  3. The reduction of microhardness for the modified cements is about a half. What are the microhardness requirements for such cements and does the manufactured materials meet these requirements? By the way, figure numbering is wrong (lines 274-275).
  4. Cytotoxicity again. It is obvious that the dilution of the extract will have the effect of reducing the cytotoxicity. However, what is the purpose of dilution? The authors defined the "acceptable" concentration of the extract. Nevertheless, this concentration is completely different from that present in the organism. So what is the purpose of cytotoxicity test? It should be explained.
  5. Fig 8 and 9 – symbols A, AC, B etc. represents significant difference. Please explain if these differences relate to values marked with i.e. "A" or if they relate to controls.
  6. Regarding comment #2, manufactured cements are still cytotoxic. The argument that the 50% extract is not cytotoxic is inappropriate here.
  7. In its present form, the article shows the improvement of the setting time through the use of magnesium modification. However, cytotoxicity results should eliminate the material from use. In my opinion, negative results can also be published as long as a scientific (rather than utilitarian) goal is defined.
